# Characterization of the 12S rRNA Gene Sequences of the Harvester Termite *Anacanthotermes ochraceus* (Blattodea: Hodotermitidae) and Its Role as A Bioindicator of Heavy Metal Accumulation Risks in Saudi Arabia

**DOI:** 10.3390/insects10020051

**Published:** 2019-02-08

**Authors:** Reem Alajmi, Rewaida Abdel-Gaber, Noura AlOtaibi

**Affiliations:** 1Zoology Department, College of Science, King Saud University, Riyadh 11451, Saudi Arabia; 2Zoology Department, Faculty of Science, Cairo University, Cairo 12613, Egypt; 3Department of Biology, Faculty of Science, Taif University, Taif 21974, Saudi Arabia; nj99alotaibi@hotmail.com

**Keywords:** Blattodea, molecular analysis, trace metals, accumulation, biological indicators

## Abstract

Termites are social insects of economic importance that have a worldwide distribution. Identifying termite species has traditionally relied on morphometric characters. Recently, several mitochondrial genes have been used as genetic markers to determine the correlation between different species. Heavy metal accumulation causes serious health problems in humans and animals. Being involved in the food chain, insects are used as bioindicators of heavy metals. In the present study, 100 termite individuals of *Anacanthotermes ochraceus* were collected from two Saudi Arabian localities with different geoclimatic conditions (Riyadh and Taif). These individuals were subjected to morphological identification followed by molecular analysis using mitochondrial 12S rRNA gene sequence, thus confirming the morphological identification of *A. ochraceus*. Furthermore, a phylogenetic analysis was conducted to determine the genetic relationship between the acquired species and other termite species with sequences previously submitted in the GenBank database. Several heavy metals including Ca, Al, Mg, Zn, Fe, Cu, Mn, Ba, Cr, Co, Be, Ni, V, Pb, Cd, and Mo were measured in both collected termites and soil samples from both study sites. All examined samples (termite and soil) showed high concentrations of metals with different concentrations and ratios. Generally, most measured metals had a significantly high concentration in soil and termites at Taif, except for Ca, Cd, Co, Cr, Cu, Mg, and Ni showing significantly high concentrations at Riyadh. Furthermore, termites accumulated higher amounts of heavy metals than the soil at both locations. The mean concentrations of the measured metals in soil samples were found to be in the descending order Ca ˃ Al ˃ Mg ˃ Zn ˃ Fe ˃ Cu ˃ Mn ˃ Ba ˃ Cr ˃ Co ˃ Be ˃ Ni ˃ V ˃ Pb ˃ Cd ˃ Mo, while it was Ca ˃ Mg ˃ Al ˃ Fe ˃ Zn ˃ Cu ˃ Mn ˃ Be ˃ Ba ˃ Pb ˃ Cr ˃ V ˃ Ni ˃ Cd ˃ Mo ˃ Co in termite specimens. The mean concentrations of the studied metals were determined in the soil and termite specimens at both locations. In addition, the contamination factor, pollution load index (PLI) and degree of contamination were calculated for all studied metals in different samples, indicating that both studied sites were polluted. However, Taif showed a significantly higher degree of pollution. Thus, the accurate identification of economically important insects, such as termites, is of crucial importance to plan for appropriate control strategies. In addition, termites are a good bioindicator to study land pollution.

## 1. Introduction

Termites are an interesting group of social insects that are widely distributed in low-land tropical ecosystems, where they can make up to 95% of the soil insect biomass [1]. The order Blattodea has been classified into 280 genera and over 2600 species within 12 families and 14 subfamilies [2]. The isopteran species are phylogenetically separated into lower termites (Mastotermitidae, Kalotermitidae, Hodotermitidae, Termopsidae, Rhinotermitidae, and Serritermitidae) and higher termites (Termitidae) [3,4]. *Anacanthotermes* is an Old-World genus of harvester termites in the Hodotermitidae. They are distributed in the deserts and semi-deserts of North Africa, the Middle East, and southwest Asia, including Baluchistan and southern India [5]. 

Lack of taxonomic understanding has been a major impediment to the study and management of termites [6]. The identification of termite species is challenging because of the ambiguity in their morphological characters and crypto-biotic social structure [7,8]. Molecular tools have become ways to complement the value of morphological taxonomy in addition to understand the evolutionary relationships among species. Molecular taxonomy based on mitochondrial DNA has proved to be an efficient alternative to the identification of species and determines their phylogenetic relationships [9,10]. DNA sequences of the mitochondrial genes of cytochrome oxidase subunit II (COX II) and the large (16S) and small (12S) subunits of ribosomal RNA (r RNA) have been used extensively for molecular diagnostics and to conduct comparative genetic analyses to study taxonomy, gene flow, colony characterization, and genetic variations [4,11]. Genetic diversity in subterranean termites was also studied using the random amplified polymorphic DNA (RAPD) markers [12]. 

Natural ecosystems all over the world have been adversely affected by human interventions [13]. Modern farming, industrialization, and increased vehicular use have led to high concentrations of heavy metals such as Pb, Ni, Cr, Cd, Al, Hg, and Zn in the environment [14]. These toxic heavy metals are regularly moved into air, water, and soil, thereby becoming a part of the natural biogeochemical cycle [15]. Insects have a strong relationship with ecology and have been frequently used as bioindicators [16]. The acute and chronic effects of heavy metals on various insects are reported frequently in the form of growth inhibition [17], developmental abnormalities [18], reduced reproduction [19], decreased hatchability [20], and alterations of respiratory and metabolic processes [21]. Insects are the agents of soil denudation owing to their feeding on soil as well as the construction of their nests in soil [22], and in the process accumulate large amounts of heavy metals in their bodies, resulting in physiological toxicity [23]. Furthermore, heavy metals could be moved to organism’s higher position in the food chains resulting in bioaccumulation and eventual toxicity [24].

Therefore, the present study aimed to determine the morphological and molecular characterizations of the harvester termite *Anacanthotermes ochraceus* to determine its taxonomic position within the family Hodotermitidae and to assess whether this hodotermitid species could be used as a candidate bioindicator of soil heavy metals.

## 2. Materials and Methods

### 2.1. Study Area

Two distinct natural habitats in Saudi Arabia were selected for the current investigation: Site I at An-Nazim District, Eastern region of Riyadh City (24°41′15.83″ N, 46°43′18.66″ E) and Site II at Arfaa District, Northern region of Taif City (21°16′36.2892″ N, 40°22′28.1172″ E). Both sites are considered as polluted areas. Site I was characterized by the presence of sandy hills, some desert plants with high temperature, illegal dumping of raw sewage into public use areas, road traffic, and industrial quarters, whereas, site II was characterized by a sandy area with small rocks including desert plants with high temperature, a cement plant, and vehicle maintenance workshops. 

### 2.2. Samples Collection and Examination

A total of 100 individuals for all castes of the harvester termite *Anacanthotermes ochraceus* were randomly collected from indexed wood and sand of two distinct natural habitats in Saudi Arabia. After collection, the samples were transported immediately to the Laboratory of Entomology Research, College of Science, King Saud University, Riyadh, Saudi Arabia, for further investigations. The termite specimens were collected in plastic vials with 70% ethyl alcohol; several were studied under stereoscopic binocular microscope. The collected termites were identified based on their external morphological characteristics, such as shape and size of the head, labrum, mandibles, pronotum, postmentum, and position of the tooth and antennal segments, etc., according to Badawi et al. [25].

### 2.3. Molecular Analysis

DNA was extracted from the ethanol-preserved specimens using the DNeasy Tissue Kit (Qiagen, Hilden, Germany) in accordance with the manufacturer’s instructions. The PCR amplification of a 5′ portion of the 12S rRNA was carried out using the termite-specific primers 12S-F (SR-J-14199) 5′-TAC TAT GTT ACG ACT TAT-3′ and 12S-R (SR-N-14594) 5′-AAA CTA GGA TTA GAT ACC C-3’ designed by Murthy et al. [26]. Each reaction mixture of 25 μL consisted of 2.5 μL of 10× PCR buffer, 2.0 μL MgCl_2_ (2.5 mM), 0.2 μL dNTPs (200 μM), 1 μL of Taq Polymerase (1U/μL), 1 μL of each primer, 1 μL of extracted DNA, and 16.3 μL of distilled H_2_O. The PCR program included an initial denaturation at 94 °C for 15 min, followed by 45 cycles of denaturation at 94 °C for 45 s, annealing at 41 °C for 45 s, extension at 72 °C for 45 s, and a final extension at 72 °C for 10 min. The amplified DNA fragments were separated on 2.5% agarose gel in 1× Tris-acetate-EDTA (TAE) (stained with 1% ethidium bromide) with a DNA ladder (100 bp) (Solis Biodyne, Tartu, Estonia). PCR products were visualized on a UV trans-illuminator and photographed using a gel documentation system (Image Analyzer, Malvern, UK). The PCR products were purified via Montage PCR96 Cleanup Kit (Milliopore, Darmstadt, Germany) and then sequenced using ABI BigDye 3.1® Sequencing Kit on an ABI 310 Automated DNA Sequencer (Applied Biosystems, Midland, ON, Canada). Reading frames were identified using the basic local alignment search tool (BLAST) searches (blastx) [27] as implemented by the National Center for Biotechnology Information (NCBI) website. The sequences of all termite species were aligned and compared with the species obtained from PUBMED using CLUSTAL-X multiple sequence alignment [28] (Table 1). Gaps were treated as missing characters in all sequence alignments. A phylogenetic tree was tested by applying a bootstrap test [29]. A phylogenetic tree was constructed using the character-based maximum-likelihood method based on the Tamura-Nei model [30]. The nodal support values among branches were assessed through a bootstrap analysis with 1000 replicates. The constructed phylogenetic tree was visualized using MEGA ver.6 [31]. The tree was drawn to scale, with branch lengths in the same units as those of the evolutionary distances used to infer the phylogenetic tree. 

### 2.4. Chemical Analysis for Measuring Heavy Metals 

The metal content of the whole termite bodies and soil samples was quantified. Termite tissues were cleaned by washing with double distilled H_2_O. Soil samples were collected within 1 m from each termite colony in two sites under investigation. Soil samples were obtained by removing a 15 × 15 cm portion of soil to a depth approximately 10 cm, and then crushed with a wooden pestle and mortar to pass through a 2 mm pore sieve. Termite and soil samples were dried in oven at 105 °C for 6–12 h, and then weighed using a microbalance. The metal content was determined by digesting 0.5 g of the oven-dried material in 5 mL HNO_3_ at 550 °C for 6–12 h, after which 2 mL of HClO_4_ was added and the volume adjusted to 10 mL with deionized H_2_O. Concentration of elements was performed using an inductively coupled plasma-optical emission spectroscopy (ICP-OES Varian 720/730-ES Series Spectrometers). All metals were determined and expressed as mg/g dry weight. The absorption wavelengths and detection limits for the heavy metals were 396.152 nm and 0.12 ppm for Al, 313.042 nm and 0.05 ppm for Be, 455.403 nm and 0.16 ppm for Ba, 315.887 nm and 1.05 ppm for Ca, 214.439 nm and 0.04 ppm for Cd, 238.892 nm and 0.02 ppm for Co, 267.716 nm and 0.04 ppm for Cr, 327.395 nm and 0.05 ppm for Cu, 238.204 nm and 0.01 ppm for Fe, 279.553 nm and 0.01 ppm for Mg, 257.610 nm and 0.05 ppm for Mn, 202.032 nm and 0.05 ppm for Mo, 231.604 nm and 0.001 ppm for Ni, 220.353 nm and 0.13 ppm for Pb, 306.772 nm and 0.02 ppm for V, and 213.857 nm and 0.04 ppm for Zn, respectively.

The contamination factor (CF) for soil was determined according to Håkanson [32] as follows: CF = C_Heavy metal_/C_Background_, the “CF < 1” values indicate low contamination; the “1 < CF < 3” values indicate moderate contamination; the “3 < CF < 6” values indicate considerable contamination; and the “CF ˃ 6” values indicate very high contamination. The pollution load index (PLI) was used in order to compare the total content of heavy metals at the studied sites according to Usero et al. [33] as follows: PLI = (CF1* CF2* CF3* …… * CF_n_)^1/n^. N is the number of metals; CF is the contamination factor; PLI = 0 indicates perfection; PLI=1 indicates the presence of only baseline levels of pollutants; PLI ˃ 1 indicates the progressive deterioration of the site quality. To determine the sum of all contamination factors for a given site, the degree of contamination (DC) was calculated as follows: DC = $\overset{n}{\underset{i=1}{\sum }}\mathrm{CFi}$. CF: represents a single contamination factor; n: indicates the count of the elements present at a site; DC<n indicates low degree of contamination; n ≤ DC < 2n indicates moderate degree of contamination; 2n ≤ DC < 4n indicates considerable degree of contamination; DC > 4n indicates very high degree of contamination. To express the accumulation capacity of termites, the ratios of C_[termite]_/C_[soil]_ (Bioconcentration factors, BCF) (bioaccumulation was considered, when the values of BCF ≥ 1) were calculated according to Sures et al. [34]. Additionally, the concentration ratio between the termites in Site I to those in Site II was determined.

### 2.5. Statistical Analysis 

All data obtained from the experiment were presented as mean ± SE and subjected to the one-way analysis of variance test (ANOVA) from which the significant relationships between the accumulation of heavy metals in termites and their inhabited sites were established. All statistical procedures were performed with the SPSS statistics 16.0 software (SPSS Inc., Chicago, IL, USA). The differences were considered significant, when the *p*-values were ≤0.01 and ≤0.05. In addition, the Pearson’s correlation coefficients (r) were calculated to check the possible relationship/interactions between the heavy metal concentrations in different localities and their samples.

## 3. Results

The termite samples were collected from two different sites (I and II) in Saudi Arabia. *A. ochraceus* soldiers were identified as the most abundant species across the two study sites. It was morphologically characterized by a shiny body covered with tiny hairs, oblong head capsule (mean length 0.4 mm and width 0.3 mm) and curved posterior margin. Eyes were black and small. Mandibles were well serrated and asymmetric in length (the left one was 0.5 mm and the right one was 0.3 mm). Antenna consisted of 25 segments. Thorax and legs were yellowish and developed. The pronotum was broad and the abdomen was white in color.

### 3.1. Molecular Analysis

A sequence of 407 bp was deposited in GenBank (gb|MG197798.1) with 34.2% GC content for the partial 12S rRNA gene sequences of the present hodotermitid species. A pairwise comparison of the isolated genomic sequence of the present termite species with a range of other isopteran species and genotypes revealed a unique genetic sequence. Calculating the percentage of identity between this novel genetic sequence and the other sequences retrieved from the GenBank demonstrated a high degree of similarity up to 80.0%. A comparison of the nucleotide sequences and divergence showed that 12S rRNA of the present species revealed the highest blast scores with lower divergence values for *A. ochraceus* (gb|DQ441629.1), *Microhodotermes viator* (gb|EU253701.1, DQ441739.1), *Hodotermopsis sjostedti* (gb|DQ441708.1), *Hodotermes mossambicus* (gb|MF554678.1), *Zootermopsis angusticollis* (gb|DQ441841.1), *Mastotermes darwiniensis* (gb|EF623330.1), *Archotermopsis wroughtoni* (gb|DQ441642.1), and *Porotermes quadricollis* (gb|DQ441794.1) (Figure 1). For estimating maximum likelihood (ML) values, a tree topology was automatically computed. The constructed dendrogram supports the suprafamilial termite lineages of Euisoptera and Neoisoptera, a later derived Kalotermitidae as a sister group of the Neoisoptera and a monophyletic clade of the dampwood (Stolotermitidae, Archotermopsidae) and harvester termites (Hodotermitidae) with strong support values (Figure 2). Within the Neoisoptera, Rhinotermitidae is consistently paraphyletic with respect to Termitidae, and Rhinotermitinae (represented by *Schedorhinotermes*) acts as a sister group to the remaining rhinotermitids plus termitids. Monotypic family Serritermitidae is nested within Rhinotermitidae. Serritermitidae is recovered as sister group to Rhinotermitidae ‏and Termitidae. The resolution within Termitidae consistently supported Mastotermitidae as a sister group of the remaining termites (=Euisoptera), while there is support for the monophyletic Termitinae (i.e., *Drepanotermes* and *Macrognathotermes*) to the exclusion of *Nasutitermes* (Nasutitermitinae). The nodal supports were very high for the family-level relationships in termites. In addition, Hodotermitidae was recovered as a monophyletic group with the three genera of *Anacanthotermes*, *Hodotermes*, and *Microhodotermes* having nodal strong support values. The present sequence in conjunction with the existing data investigates the placement of this hodotermitid species within the infraorder Isoptera within order Blattodea. It was shown that the present species is deeply embedded in the genus *Anacanthotermes* in the same taxon with the previously described *A. ochraceus*. 

### 3.2. Chemical Analysis

The metal concentrations in the soil and termites collected from sites (I) and (II) in Saudi Arabia are shown in Table 2. All examined samples contained high concentrations of metals. Mean concentrations of the measured metals in soil samples were: Ca ˃ Al ˃ Mg ˃ Zn ˃ Fe ˃ Cu ˃ Mn ˃ Ba ˃ Cr ˃ Co ˃ Be ˃ Ni ˃ V ˃ Pb ˃ Cd ˃ Mo. Generally, the most measured metals had a significantly high concentration at Site II, except for Ca, Cd, Co, Cr, Cu, Mg, and Ni, which had significantly high concentration at Site I. Some toxic elements (Cd, Co, and Mo) were detected at concentrations lower than 0.5 mg/g. Moderate concentrations were recorded for Be, Ba, Cr, Cu, Mn, Ni, Pb, and V, whereas the essential elements of Al, Ca, Fe, Mg, and Zn were found exceed 50 mg/g. 

Termites accumulated high amounts of heavy metals (Table 2). The order of metal concentrations in termites was Ca ˃ Mg ˃ Al ˃ Fe ˃ Zn ˃ Cu ˃ Mn ˃ Be ˃ Ba ˃ Pb ˃ Cr ˃ V ˃ Ni ˃ Cd ˃ Mo ˃ Co. The highest concentrations of Ba, Al, Cr, Fe, Ni, V, and Zn were found in the termites inhabiting Site II. While, the elements Be, Ca, Cu, Mg, Mn, and Pb were found to accumulate to a significantly high degree in Site I. Among metals, the concentration of calcium was found to be the highest, whereas that of Co was the lowest in both soil samples and termites. When comparing the termite and soil metal content, the concentration of a metal within the termite bodies of Site II was always significantly correlated with the amount in the soil, except for Co, which had the highest concentration in the soil samples of Site I as shown in Table 2. The biggest difference involved Al, Ca, Cu, Fe, and Mg, where more than four-fold difference was observed in both soil samples and termites.

The CF of heavy metals in the soil samples of Site I (Table 3) represented two categories of contamination as moderate CF (1 ˂ CF ˂ 3) varying from 1.1946 to 2.6501 for Be, Cd, and Pb; and high CF (CF ˃ 3) varying from 4.7661 to 66530.3 for Al, Ba, Ca, Co, Cr, Cu, Fe, Mg, Mn, Mo, Ni, V, and Zn. The sequence of CF for heavy metals in the soil of Riyadh in this region was: Mg > Fe > Al > Zn > Ca > Ni > Cu > Mn > Co > Cr > Ba > V > Mo > Cd > Be > Pb. This sequence shows that Mg is the most abundant metal, whereas Pb showed the lowest appearance. In addition, Site II represented two categories of contamination as moderate CF (1 ˂ CF ˂ 3) varying from 1.3242 to 1.9701 for Be, Cd, Co, and Pb; and high CF (CF ˃ 3) varying from 5.4052 to 3232.6 for Al, Ba, Ca, Cr, Cu, Fe, Mg, Mn, Mo, Ni, V, and Zn. The descending sequence of CF for metals in the soil of Site II was: Fe > Al > Mg > Zn > Ca > Ni > Mn > Cu > Ba > Mo > V > Cr > Cd > Be > Pb > Co, showing that Fe is the most abundant metal; whereas Co showed the lowest appearance.

The PLI and degree of contamination (DC) of heavy metals in the soil samples from the two studied sites are shown in Table 3. The PLI of Site (I) ranged between 0.0192 and 1.7578 and showed a progressive deterioration of site quality. The contamination degree in Site I indicates a considerable degree of pollution. The PLI of Site II ranged between 0.0347 and 2.6525.

According to the bioconcentration factors, termites showed the highest accumulation rates up to 50.377 for all recorded elements as shown in Table 4. The comparison of element concentration in termites between the two sites (I, II) revealed a clear difference. Accordingly, the C_[*A. ochraceus* Site I]_/C_[*A. ochraceus* Site II]_ ratios for the most essential elements were higher than 0.5, with the following order Ca ˃ Be ˃ Cu ˃ Mo ˃ Cd ˃ Mg ˃ Pb ˃ Ni ˃ Ba ˃ Cr ˃ V ˃ Zn ˃ Mn ˃ Fe ˃ Co ˃ Al.

The assessment of the overall relationship between the collected samples with heavy metal concentrations showed that most elements had weak to very strong correlation. Table 5 shows significant positive correlation of Al with Mo and Pb (r = 0.999 and 1.000, respectively) and a negative correlation with Cr (r = −0.572). The metal Be showed a significant positive correlation with Fe (r = 1.000) and a negative correlation with Cr (r = −0.348). Ba showed a significant positive correlation with Ca (r = 1.000) and a negative correlation with Cr (r = −0.244). Both Ca and Cd correlated negatively with Cr (r = −0.228 and −0.645, respectively). Co correlated negatively with Mg and V (r = −0.446 and −0.483, respectively). Cr showed a negative correlation with Cu, Fe, Mg, Mn, Mo, Ni, and Pb (r = −0.434, −0.340, −0.997, −0.503, −0.541, −0.499, and −0.572, respectively) and a positive correlation with V (r = 0.999). Cu showed significant positive correlation with Mn and Ni (r = 0.997). Mg showed significant positive correlation with V (r = 0.999), and Mn with Mo and Ni (r = 0.999 and 1.000, respectively). Mo showed significant positive correlation with Ni and Pb (r = 0.999), whereas, V correlated negatively with Zn (r = −0.307). 

Table 6 shows a significant positive correlation between Al and Cu (r = 0.998) and a negative correlation between Al and Cd (r = −0.875). Be had a significant positive correlation with Ca and Fe (r = 1.000 and 0.999, respectively) and negative correlation with Cd (r = −0.971). Ba and Ca correlated negatively with Cd (r = −0.219 and −0.975, respectively), and Cd correlated negatively with Co, Cr, Cu, Fe, Mg, Mn, Mo, Ni, Pb, and V (r = −0.998, −0.383, −0.905, −0.959, −0.996, −0.998, −1.000, −0.999, −0.993, and −0.832, respectively). Co correlated positively with Mg, Mo, Ni, and Pb (r = 1.000, 0.997, 1.000, and 0.998, respectively). Mg correlated significantly positively with Ni and Pb (r = 0.999 and 1.000, respectively), and Mn correlated Mo (r = 0.999). Mo correlated significantly positively with Ni (r = 0.998). Ni correlated positively with Pb (r = 0.998) and negatively with Zn (r = −0.307). 

Table 7 shows a significantly positive correlation of Al with Cu and Zn (r = 1.000). Be correlated positively with Cd and Mn (r = 0.999 and 1.000, respectively) and negatively with Cr and Ni (r = −0.469, −1.000; respectively). Ba correlated negatively with Cr and Ni (r = −0.607 and -0.984, respectively). Ca correlated positively with Mg, Pb, and V (r = 0.998, 0.998, and 0.999, respectively) and negatively with Cr and Ni (r = −0.307 and −0.987, respectively). Cd correlated positively with Mg and Mn (r = 0.998 and 0.999, respectively) and negatively with Cr and Ni (r = −0.435 and −1.000, respectively). Cr correlated negatively with Co, Mg, Mn, Pb, and V (r = −0.709, −0.372, −0.471, −0.251, and −0.256, respectively). Fe correlated significant positively with Mo (r = 1.000) and negatively with Ni (r = -0.635). A negative correlation was found between Ni and Cu, Co, Mg, Mn, Mo, Pb, and V (r = −0.840, −0.951, −0.996, −1.000, −0.618, −0.976, and −0.977, respectively) and a significant positive correlation between Pb and V (r = 1.000). A significant positive correlation was found between Zn and all other elements (r = 1.000).

Table 8 shows that Al correlated negatively with Fe (r = -0.173) and Be correlated significantly positively with Cd (r = 1.000). Ba correlated negatively with Fe and Ni (r = −0.135 and −0.347, respectively) and Ca correlated positively with Cu, Mg, Mn, and Mo (r = 1.000, 1.000, 1.000, and 0.999, respectively) and negatively with Zn (r = −0.079). Co correlated positively with Mo (r = 0.998). A significant negative correlation existed between Cr and Fe (r = −0.999). Cu correlated positively with Mg, Mn, and Mo (r = 1.000, 1.000, and 0.999, respectively) and negatively with Fe (r = −0.838). Fe correlated negatively with Mg, Mn, Mo, and Ni (r = −0.849, −0.839, −0.818, and −0.883, respectively). Mg correlated positively with Mn and Mo (r = 1.000 and 0.998, respectively). A positive correlation existed between Mn and Mo (r = 0.999). Pb correlated negatively with Al, Be, Ca, Cd, Co, Cr, and Cu (r = −0.519, −0.249, −0.436, −0.247, −0.331, −0.826, and −0.422, respectively). V correlates negatively with Al, Be, Ca, Cd, Co, and Cu (r = −0.429, −0.148, −0.341, −0.146, −0.232, and −0.327, respectively). Zn showed a negative correlation with Al, Ca, Cr, Cu, Mg, Mn, Mo, Ni, Pb, and V (r = −0.173, −0.079, −0.564, −0.063, −0.872, −0.935, −0.918, −0.888, −0.858, and −0.530, respectively). Other relations among elements showed low or no significant correlations.

## 4. Discussion

Termites are an interesting and economically important group of social insects, scientifically classified in the order Blattodea and are widely distributed in tropical and subtropical regions [35]. They have a highly evolved social organization and hierarchical structure [36]. They also live in colonies of various sizes containing several different castes which may include workers, soldiers, nymphs, and larvae [37]. In termites, the soldier and worker mandibles play crucial roles in defense and feeding functions [38]. Their morphology might vary from one genus or even from one species to another and is therefore commonly used for the systematic description of termite species.

According to the present results, *A. ochraceus* was naturally recorded in both localities under investigation; this is consistent with data reported by Ahmad [39] and Roonwal and Rathore [40], whom stated that species of the genus *Anacanthotermes* belonging to the family Hodotermitidae are characterized as the termites of Blattodea. Furthermore, since the beginning of the century, termites have proved to be of great interest in the study of phylogenetic relations between different taxa, as illustrated by the previous studies of Lynch [41], Jenkins et al. [42], and Szalansky et al. [43]. In the present study, the 12S rRNA region of the recovered termite species was amplified using the species-specific primers 12S-F/12S-R, designed previously by Murthy et al. [26]. It is apparent that the present phylogenetic tree strongly supported the taxonomic groups of termites into four major lineages—the Mastotermitidae, the Stolotermitidae plus Hodotermitidae plus Archotermopsidae (SHA clade), the Kalotermitidae, and the Neoisoptera (=Rhinotermitidae plus Termitidae). These results agree with data obtained by Cameron et al. [44] and Legendre et al. [45]. The phylogenetic analysis of termite evolution described here identified the Mastotermitidae as the sister group of the remaining extant termites (=Euisoptera). This result agreed with the result of Nalepa and Lenz [46], who demonstrated that the sister-grouping of Mastotermitidae plus Euisoptera is one of the few well-supported clades that are consistent with the morphological, biological, and genomic data. The second lineage of the SHA clade was also reported by Legendre et al. [47]. All these families have been previously considered as a single family, Hodotermitidae [39]; however, the diverse life-histories found in the group have supported their division into multiple families. In the present study, Hodotermitidae was considered a monophyly in origin; this result is consistent with the studies of Thompson et al. [48] and Inward et al. [49], who found that Hodotermitidae was a monophyletic family nested within the paraphyletic groups of Termopsidae and Serritermitidae. The Archotermopsidae, however, was found to be a non-monophyletic group in all studies, where its monophyly was testable, including the one in which it was proposed [50], and it might, therefore, be taxonomically more conservative to revert to a single family, Hodotermitidae. Stolotermitidae and Archotermopsidae have quite contrasting biology [51]. They were long considered a single family, the Termopsidae; however, their disjoint distribution stolotermitids occur in the southern hemisphere and archotermopsids in the northern hemisphere combined with their frequent non-monophyly in the phylogenetic analyses led Engel et al. [50] to raise the former to family status, Stolotermitidae, and to propose the Archotermopsidae as a new name for the northern termopsids as the type genus for this family was extinct and not closely related to the extant genera. The third major lineage was identified as the drywood termites, Kalotermidae. In almost all previous phylogenetic hypotheses, the Kalotermitids were monophyletic (paraphyletic with respect to Neoisoptera in Donovan et al. [52]). In the present study, their position seemed to be well supported as a sister group to Neoisoptera, as reported by Kambhampati and Eggleton [3], Inward et al. [49], Legendre et al. [47], Ware et al. [53], Lo and Eggleton [54], and Cameron et al. [44], who supported their position as the second diverging lineage after *Mastotermes* within the termites, and this presents a new hypothesis for the placement of this family. In addition, Ahmad [39] mentioned about the Kalotermitidae that "the imago-worker mandibles are essentially the same as in Mastotermitidae" and clustered Kalotermitidae and Mastotermes based on those mandibles.

The Neoisoptera has been universally supported by the previous and present phylogenetic analyses. The Rhinotermitidae was paraphyletic on our tree, just as has been found by Kambhampati et al. [55], Donovan et al. [52], Thompson et al. [48], and Lo et al. [56], who reported the same data with support for only two major clades: the early branching Rhinotermitinae and the late branching [Coptotermitinae plus Heterotermitinae]. Another possible explanation for the non-monophyly of Rhinotermitidae was long hypothesized because of the high behavioral, developmental and morphological diversity of this group as stated by Grassé [57]. In the present study, the monophyly of the Termitidae is well established and supported by our topology; this finding agreed with the results of Inward et al. [49]. Another family, which might also warrant reclassification, is the enigmatic Brazilian Serritermitidae, which has been recovered consistently within the part of the (paraphyletic) Rhinotermitidae in our analysis, rather than a sister group to the Rhinotermitidae plus Termitidae as suggested by Inward et al. [49]. Our topology recognized the species-rich pan-tropical genera *Heterotermes* and *Nasutitermes* presently within Blattodea. Our analysis suggests that the first one was monophyletic, and more extensive taxon sampling will be required for the determination of the second one; this was consistent with the opinion of Inward et al. [49]. In addition, the genomic sequences of the rRNA gene for the present *Anacanthotermes* species demonstrated the monophyly of the family Hodotermitidae and supported the taxonomic position of the present termite species, which is embedded deeply in the genus *Anacanthotermes* with a close relationship with the previously described *A. ochraceus* as a more related sister taxon. This data coincided with previous studies [39,40,47,48,49].

Metal accumulation around the world has increased with the onset of urbanization and industrialization [58]. Our study demonstrated that the evaluation of heavy metal concentration in soil and termites leads to the bio-assessment of heavy metal pollution. Termites accumulated high concentrations of metals more than soil in both localities under investigation, except for Co (Site I), and Mo (Site II). The concentrations of these contaminants were found in the hazardous levels that exceeded the permissible limits for mineral soil recommended by the standards of the Egyptian High Committee of Water [59], Saudi Standards Metrology and Quality Organization [60], and United States Environmental Protection Agency [61]. These results were consistent with Gadd [62], Bååth et al. [63], Pennanen et al. [64,65], and Zafar et al. [66], who stated that the elevated heavy metal concentrations in soil are one of the stress factors exerting a selection pressure on soil microorganisms. Similar results were observed by Jop [67] in several indigenous organisms captured along a metal pollution gradient such as aquatic insects, which showed a gradual increase in metal concentrations in various metal-sensitive and metal-detoxified sub-cellular fractions with increasing environmental metal levels. In addition, Giguère et al. [68] stated that these organisms were able to protect themselves against metal toxicity by increasing the proportion of metals in non-toxic forms. The body metal concentrations in termites originated from Site II were higher than in those originated from Site I; this result was consistent with the findings of Maavara et al. [69] and Rabitsch [70], who reported that metal accumulation in termites was studied in relation to the differences between casts, interspecific variation tissue specific accumulation and time-related effects. Furthermore, Rabitsch [71] suggested that the high metal accumulation capacity of termites is accomplished through active metal excretion. In the present study, Ca was accumulated in significantly higher concentration than other elements, while the least accumulated ones were Co and Mo. Other elements such as Cu, Fe, and Zn, are considered important and essential nutrients for organisms. However, they could be very toxic at high concentration for cells and cause metabolic damage as stated by Hackman [72], Warrington [17], Gadd [62], Bagatto and Shorthouse [73], Lu et al. [74], Zan et al. [75], and Cheruiyot et al. [76]. In addition, Roth-Holzapfel [77], Barajas-Aceves et al. [78] and Zafar et al. [66], who reported that the accumulation factor for grasshopper was Cd > Hg > Pb, indicating that different metals have different affinities leading to bioaccumulation in different organisms. Moreover, Bose and Bhattacharyya [79] and Toth et al. [80] reported that the high bioaccumulation values might be related to heavy metal-binding capacity of tissues, available metals, interactions between physicochemical parameters, and different species grown in these soils. 

In the present study, the high ratios of CF, PLI, and DC were recorded in both studied sites, with Site II showing higher values than Site I because of the characteristic features of the studied area, the possible explanation of these elevations according to Grzesiak and Sieradzki [81] is that the emission of pollutants from the petroleum industries into the environment has been considerably affecting the flora and fauna, which accumulate large amounts of associated heavy metals. Correlation analyses assist to reveal the relationship of termites with the concentration of elements in soil. The current investigation recorded a significant direct relationship between the presence of termites on the concentrations of Al, Cu, Zn, Be, Cd, Mn, Ca, Mg, Pb, V, and Mo, while, a significant indirect correlation existed for Ba, Cr, Ni, Co, and Fe. This result agreed with Rogival et al. [82] reported linear relationships between the total metal concentrations and the bioavailable fraction in the soil and levels in wood mice tissues. In addition, Flannagan et al. [83] cited that a possible explanation for this correlation is that the inhibition of AChE activity in the heads of stonefly nymphs was linearly correlated with the concentrations in nymphs poisoned by fenitrothion. Furthermore, Ibrahim et al. [84] and Iyaka and Kakulu [85] demonstrated similar results for stoneflies and chironomids exposed to several organophosphorus compounds.

## 5. Conclusions

Therefore, it could be concluded that the molecular studies provide more information for accurate taxonomic identification along the species level of the harvester termite *A. ochraceus* inhabiting Saudi Arabia. The dominance of the terrestrial insects in these ecosystems and the differential responsiveness of species to environmental stressors have led to their extensive use as ecological indicators worldwide. In addition, advanced future studies are recommended for elucidating the effect of heavy metals on the development of different harvester termites.

## Figures and Tables

**Figure 1 insects-10-00051-f001:**
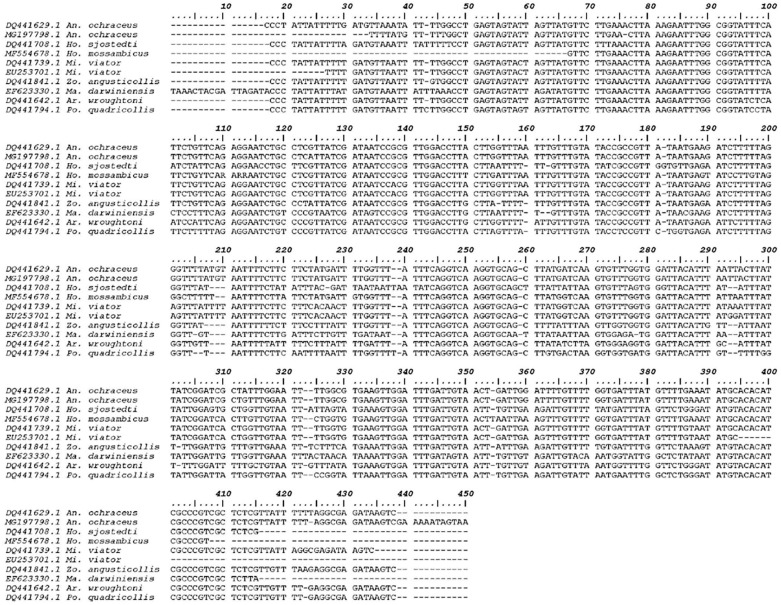
Sequence alignment of the 12S rRNA of *A. ochraceus* with the most related species (Only variable sites are shown. Dots represent the bases identical to those of the first sequences, and dashes indicate gaps).

**Figure 2 insects-10-00051-f002:**
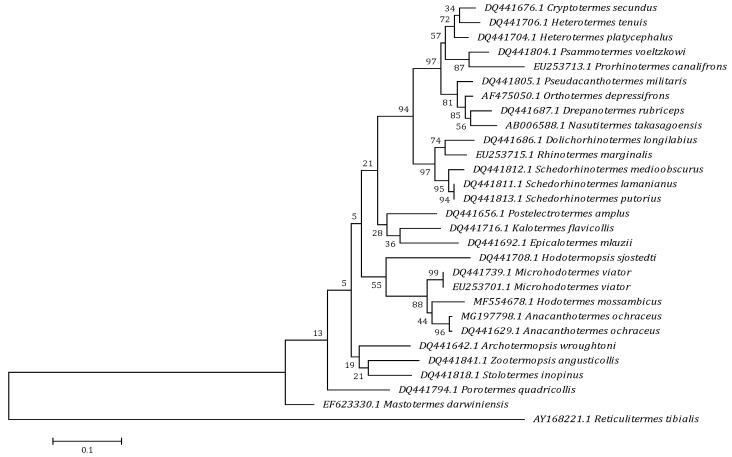
Molecular phylogenetic analysis by maximum likelihood based on the Tamura-Nei model. The tree with the highest log likelihood −2940.2934) is shown. The percentage of trees in which the associated taxa clustered together is shown next to the branches. The initial tree(s) for the heuristic search were obtained automatically by applying the Neighbor-Join and BioNJ algorithms to a matrix of pairwise distances estimated using the maximum composite likelihood (MCL) approach, and then selecting the topology with superior log likelihood value.

**Table 1 insects-10-00051-t001:** Termitid species used in the phylogenetic analysis of the present *A. ochraceus.*

Parasite Species	Order/Family	Source	Accession No.	Sequence Length (bp)	Percent Identity (%)
*Cryptotermes secundus*	*Blattodea/Kalotermitidae*	GenBank	DQ441676.1	407	80
*Heterotermes tenuis*	*Blattodea/Rhinotermitidae*	GenBank	DQ441706.1	416	81
*Heterotermes platycephalus*	*Blattodea/Rhinotermitidae*	GenBank	DQ441704.1	413	81
*Psammotermes voeltzkowi*	*Blattodea/Rhinotermitidae*	GenBank	DQ441804.1	417	83
*Prorhinotermes canalifrons*	*Blattodea/Rhinotermitidae*	GenBank	EU253713.1	365	83
*Pseudacanthotermes militaris*	*Blattodea/Termitidae*	GenBank	DQ441805.1	413	84
*Orthotermes depressifrons*	*Blattodea/Termitidae*	GenBank	AF475050.1	438	82
*Drepanotermes rubriceps*	*Blattodea/Termitidae*	GenBank	DQ441687.1	412	82
*Nasutitermes takasagoensis*	*Blattodea/Termitidae*	GenBank	AB006588.1	404	82
*Dolichorhinotermes longilabius*	*Blattodea/Rhinotermitidae*	GenBank	DQ441686.1	415	83
*Rhinotermes marginalis*	*Blattodea/Rhinotermitidae*	GenBank	EU253715.1	359	83
*Schedorhinotermes medioobscurus*	*Blattodea/Rhinotermitidae*	GenBank	DQ441812.1	414	83
*Schedorhinotermes lamanianus*	*Blattodea/Rhinotermitidae*	GenBank	DQ441811.1	413	83
*Schedorhinotermes putorius*	*Blattodea/Rhinotermitidae*	GenBank	DQ441813.1	392	83
*Postelectrotermes amplus*	*Blattodea/Kalotermitidae*	GenBank	DQ441656.1	407	83
*Kalotermes flavicollis*	*Blattodea/Kalotermitidae*	GenBank	DQ441716.1	407	84
*Epicalotermes mkuzii*	*Blattodea/Kalotermitidae*	GenBank	DQ441692.1	402	82
*Hodotermopsis sjostedti*	*Blattodea/Termopsidae*	GenBank	DQ441708.1	388	90
*Microhodotermes viator*	*Blattodea/Hodotermitidae*	GenBank	DQ441739.1	409	93
*Microhodotermes viator*	*Blattodea/Hodotermitidae*	GenBank	EU253701.1	360	93
*Hodotermes mossambicus*	*Blattodea/Hodotermitidae*	GenBank	MF554678.1	333	90
*Anacanthotermes ochraceus*	*Blattodea/Hodotermitidae*	GenBank	DQ441629.1	414	99
*Archotermopsis wroughtoni*	*Blattodea/Termopsidae*	GenBank	DQ441642.1	406	82
*Zootermopsis angusticollis*	*Blattodea/Termopsidae*	GenBank	DQ441841.1	405	88
*Stolotermes inopinus*	*Blattodea/Termopsidae*	GenBank	DQ441818.1	414	82
*Porotermes quadricollis*	*Blattodea/Termopsidae*	GenBank	DQ441794.1	406	87
*Mastotermes darwiniensis*	*Blattodea/Mastotermitidae*	GenBank	EF623330.1	400	85
*Reticulitermes tibialis*	*Blattodea/Rhinotermitidae*	GenBank	AY168221.1	441	83

**Table 2 insects-10-00051-t002:** Heavy metal concentrations (mg/g dry weight) in two different habitats and tissues of *A. ochraceus*.

Heavy Metal	Soil Samples from Site I (Riyadh)	Soil Samples from Site II (Taif)	*Anacanthotermes ochraceus*	
			Site I (Riyadh)	Site II (Taif)
Al	196.0533 ± 32.90069 ^c^	378.4967 ± 57.72633 ^b c^	961.9387 ± 75.84870 ^a b^	1514.1020 ± 406.63030 ^a^
Be	0.4422 ± 0.07197 ^b^	0.3940 ± 0.11393 ^b^	12.1700 ± 3.09869 ^a b^	6.3833 ± 1.88832 ^a^
Ba	2.4420 ± 1.08289 ^b^	4.1214 ± 1.45522 ^a b^	6.8607 ± 1.92336 ^a b^	8.9527 ± 2.59292 ^a^
Ca	883.5467 ± 101.56333 ^b^	455.5233 ± 88.38769 ^b^	9625.6327 ± 2672.88999 ^a^	4835.3907 ± 1277.84243 ^a b^
Cd	0.1060 ± 0.2784 ^b^	0.0847 ± 0.2207 ^b^	0.4533 ± 0.10056 ^a^	0.3529 ± 0.09159 ^a^
Co	0.519 ± 0.01354 ^b^	0.0264 ± 0.00968 ^b^	0.2793 ± 0.09498 ^b^	0.4573 ± 0.18576 ^a^
Cr	0.6619 ± 0.29671 ^b^	0.2162 ± 0.09240 ^b^	2.1513 ± 0.58658 ^a b^	2.8853 ± 0.94224 ^a^
Cu	3.0313 ± 0.84077 ^b^	2.8978 ± 0.61136 ^b^	42.7720 ± 14.69708 ^a b^	23.2357 ± 5.94914 ^a^
Fe	26.2683 ± 6.97426 ^b^	32.3260 ± 8.31177 ^b^	441.7793 ± 138.77551 ^b^	940.9813 ± 250.35349 ^a^
Mg	65.3030 ± 8.54049 ^b^	26.6207 ± 6.22629 ^b^	2009.0593 ± 523.75384 ^a^	1656.1273 ± 428.84791 ^a^
Mn	2.8454 ± 1.15156 ^b^	6.3062 ± 1.31417 ^b^	14.6213 ± 3.97113 ^a b^	11.5678 ± 2.76065 ^a^
Mo	0.0383 ± 0.01091 ^b^	0.475 ± 0.01586 ^b^	0.4327 ± 0.18244 ^a^	0.4313 ± 0.13122 ^a^
Ni	0.2883 ± 0.07573 ^b^	0.1363 ± 0.04804 ^b^	1.5987 ± 0.41954 ^a^	2.0727 ± 0.57519 ^a^
Pb	0.1553 ± 0.07672 ^b^	0.2074 ± 0.07582 ^b^	3.0707 ± 0.78007 ^a^	2.6593 ± 0.66487 ^a^
V	0.2692 ± 0.07972 ^b^	0.3537 ± 0.11612 ^b^	1.6913 ± 0.42388 ^a^	2.2733 ± 0.59009 ^a^
Zn	42.4307 ± 7.21991 ^b^	89.7970 ± 9.78533 ^a b^	124.9000 ± 31.53582 ^a b^	185.1847 ± 48.74656 ^a^

Mean values within the same row with different superscripts differ significantly at *p* ≤ 0.05.

**Table 3 insects-10-00051-t003:** Contamination factor, pollution load index, and degree of contamination of soil and *A. ochraceus* samples from two sites (I, II).

Heavy Metals	Contamination Factor (CF)		Pollution Load Index (PLI)		Degree of Contamination (DC)	
	Site I	Site II	Site I	Site II	Site I	Site II
Al	1633.7	3154.1	0.7648	1.0632	4876.3	9765.2
Be	2.2113	1.9701	0.2778	0.3923	6.2453	5.7873
Ba	15.262	25.758	0.8440	0.1756	45.785	76.256
Ca	883.35	455.52	0.4402	0.3170	2576.2	1298.6
Cd	2.6501	2.1175	0.1610	0.3159	7.9343	6.0549
Co	25.951	1.3242	0.1717	1.2919	75.987	3.6757
Cr	16.547	5.4052	0.6619	0.1899	43.765	13.657
Cu	60.626	57.956	0.1346	0.1541	175.76	169.65
Fe	2626.8	3232.6	1.6551	8.8810	7545.6	9654.7
Mg	6530.3	2662.1	1.0747	1.5895	18545.2	7644.5
Mn	56.908	126.12	0.1628	0.1495	165.87	325.54
Mo	4.7661	9.5202	0.0277	0.0347	2.0857	26.549
Ni	288.32	136.30	0.1252	0.1183	765.98	389.98
Pb	1.1946	1.5953	1.7578	0.7389	3.4598	4.2983
V	5.3842	7.0743	0.0192	0.0847	15.986	20.765
Zn	1060.7	2244.9	1.2509	2.6525	3058.4	6375.6

**Table 4 insects-10-00051-t004:** Estimates of the accumulation capacity and bioconcentration factors (BCF) C_[*A. ochraceus*]/_C_[Sites I, II]_ and ratios C_[*A. ochraceus* Site I]/_C_[*A. ochraceus* Site II]._

Heavy Metals	C_[*A. ochraceus*]/_C_[Site I]_	C_[*A. ochraceus*]/_C_[Site II]_	C_[*A. ochraceus* Site I]/_C_[*A. ochraceus* Site II]_
Al	5.1183	3.4995	0.5222
Be	28.290	19.088	1.9065
Ba	3.3319	2.5774	0.7663
Ca	10.611	8.9935	1.9906
Cd	2.7169	2.5480	1.2519
Co	6.4979	26.032	0.5249
Cr	2.5215	18.137	0.7456
Cu	22.586	7.3969	1.8408
Fe	17.825	27.608	0.5694
Mg	27.394	50.377	1.2131
Mn	4.5726	3.0160	0.6498
Mo	15.919	11.677	1.4667
Ni	5.9092	19.347	0.7713
Pb	19.774	11.263	1.1546
V	4.6089	6.1567	0.7439
Zn	2.0687	1.6891	0.6744

**Table 5 insects-10-00051-t005:** Correlation between the concentrations of heavy metals in the soil samples of Site I (Riyadh).

Variables	Concentrations of Different Heavy Metals in Site I															
	Al	Be	Ba	Ca	Cd	Co	Cr	Cu	Fe	Mg	Mn	Mo	Ni	Pb	V	Zn
**Al**	1															
**Be**	0.968	1														
**Ba**	0.935	0.994	1													
**Ca**	0.929	0.992	1.000 *	1												
**Cd**	0.996	0.941	0.899	0.891	1											
**Co**	0.406	0.623	0.704	0.716	0.321	1										
**Cr**	−0.572	−0.348	−0.244	−0.228	−0.645	0.517	1									
**Cu**	0.987	0.996	0.980	0.976	0.968	0.547	−0.434	1								
**Fe**	0.966	1.000 **	0.995	0.993	0.938	0.629	−0.340	0.995	1							
**Mg**	0.636	0.422	0.321	0.305	0.704	−0.446	−0.997	0.505	0.415	1						
**Mn**	0.997	0.985	0.961	0.959	0.985	0.480	−0.503	0.997 *	0.984	0.570	1					
**Mo**	0.999 *	0.977	0.948	0.942	0.992	0.440	−0.541	0.992	0.975	0.607	0.999 *	1				
**Ni**	0.996	0.986	0.962	0.958	0.984	0.484	−0.499	0.997 *	0.985	0.567	1.000 **	0.999 *	1			
**Pb**	1.000 **	0.968	0.935	0.929	0.996	0.407	−0.572	0.987	0.966	0.636	0.997	0.999 *	0.996	1		
**V**	0.604	0.384	0.281	0.266	0.674	−0.483	0.999 *	0.468	0.376	0.999 *	0.536	0.573	0.532	0.603	1	
**Zn**	0.574	0.761	0.827	0.836	0.497	0.981	0.343	0.697	0.766	−0.267	0.639	0.604	0.642	0.574	−0.307	1

* Correlation is significant at the 0.05 level. ** Correlation is significant at the 0.01 level.

**Table 6 insects-10-00051-t006:** Correlation between the concentrations of heavy metals in the soil samples of Site II (Taif).

Variables	Concentrations of Different Heavy Metals in Site I															
	Al	Be	Ba	Ca	Cd	Co	Cr	Cu	Fe	Mg	Mn	Mo	Ni	Pb	V	Zn
**Al**	1															
**Be**	0.965	1														
**Ba**	0.664	0.446	1													
**Ca**	0.961	1.000 *	0.432	1												
**Cd**	−0.875	−0.971	−0.219	−0.975	1											
**Co**	0.902	0.984	0.277	0.986	−0.998 *	1										
**Cr**	0.783	0.593	0.985	0.580	−0.383	0.438	1									
**Cu**	0.998 *	0.981	0.613	0.977	−0.905	0.929	0.739	1								
**Fe**	0.976	0.999*	0.487	0.998*	−0.959	0.974	0.630	0.989	1							
**Mg**	0.914	0.988	0.304	0.991	−0.996	1.000 *	0.463	0.939	0.980	1						
**Mn**	0.843	0.954	0.159	0.959	−0.998 *	0.993	0.326	0.877	0.939	0.989	1					
**Mo**	0.867	0.967	0.203	0.971	−1.000 *	0.997 *	0.368	0.898	0.954	0.995	0.999 *	1				
**Ni**	0.900	0.982	0.272	0.985	−0.999 *	1.000 **	0.433	0.927	0.973	0.999 *	0.993	0.998 *	1			
**Pb**	0.925	0.992	0.331	0.994	−0.993	0.998 *	0.487	0.948	0.985	1.000 *	0.984	0.991	0.998 *	1		
**V**	0.997	0.941	0.724	0.935	−0.832	0.864	0.831	0.989	0.955	0.877	0.796	0.823	0.861	0.891	1	
**Zn**	−0.598	−0.786	0.202	−0.796	0.911	−0.885	0.031	−0.650	−0.757	−0.872	−0.935	−0.918	−0.888	−0.858	−0.530	1

* Correlation is significant at the 0.05 level. ** Correlation is significant at the 0.01 level.

**Table 7 insects-10-00051-t007:** Correlation between the concentrations of heavy metals in *A. ochraceus* of Site I (Riyadh).

Variables	Concentrations of Different Heavy Metals in Site I															
	Al	Be	Ba	Ca	Cd	Co	Cr	Cu	Fe	Mg	Mn	Mo	Ni	Pb	V	Zn
**Al**	1															
**Be**	0.848	1														
**Ba**	0.751	0.987	1													
**Ca**	0.928	0.985	0.943	1												
**Cd**	0.868	0.999 *	0.980	0.991	1											
**Co**	0.654	0.955	0.991	0.889	0.944	1										
**Cr**	0.069	−0.469	−0.607	−0.307	−0.435	−0.709	1									
**Cu**	1.000 *	0.832	0.730	0.916	0.853	0.631	0.100	1								
**Fe**	0.943	0.623	0.487	0.751	0.653	0.365	0.398	0.952	1							
**Mg**	0.900	0.994	0.964	0.998 *	0.998 *	0.918	−0.372	0.886	0.703	1						
**Mn**	0.847	1.000 **	0.987	0.984	0.999 *	0.956	−0.471	0.831	0.622	0.994	1					
**Mo**	0.935	0.607	0.469	0.736	0.636	0.345	0.417	0.946	1.000 *	0.688	0.605	1				
**Ni**	−0.856	−1.000 **	−0.984	−0.987	−1.000 *	−0.951	0.456	−0.840	−0.635	−0.996	−1.000 *	−0.618	1			
**Pb**	0.948	0.973	0.922	0.998 *	0.981	0.861	−0.251	0.938	0.788	0.992	0.972	0.775	−0.976	1		
**V**	0.947	0.974	0.924	0.999 *	0.982	0.863	−0.256	0.936	0.785	0.993	0.973	0.772	−0.977	1.000 **	1	
**Zn**	1.000 **	1.000 **	1.000 **	1.000 **	1.000 **	1.000 **	1.000 **	1.000 **	1.000 **	1.000 **	1.000 **	1.000 **	1.000 **	1.000 **	1.000 **	1

* Correlation is significant at the 0.05 leve. ** Correlation is significant at the 0.01 level.

**Table 8 insects-10-00051-t008:** Correlation between the concentrations of heavy metals in *A. ochraceus* of Site II (Taif).

Variables	Concentrations of Different Heavy Metals in Site I															
	Al	Be	Ba	Ca	Cd	Co	Cr	Cu	Fe	Mg	Mn	Mo	Ni	Pb	V	Zn
**Al**	1															
**Be**	0.957	1														
**Ba**	0.566	0.781	1													
**Ca**	0.995	0.980	0.642	1												
**Cd**	0.956	1.000 **	0.782	0.980	1											
**Co**	0.978	0.996	0.725	0.994	0.996	1										
**Cr**	0.911	0.752	0.176	0.867	0.750	0.805	1									
**Cu**	0.994	0.983	0.654	1.000 **	0.983	0.995	0.860	1								
**Fe**	−0.893	−0.724	−0.135	−0.846	−0.723	−0.780	−0.999 *	−0.838	1							
**Mg**	0.996	0.979	0.638	1.000 **	0.979	0.993	0.870	1.000 *	−0.849	1						
**Mn**	0.994	0.983	0.652	1.000 **	0.982	0.995	0.861	1.000 **	−0.839	1.000 *	1					
**Mo**	0.989	0.989	0.680	0.999 *	0.989	0.998 *	0.841	0.999 *	−0.818	0.998 *	0.999 *	1				
**Ni**	0.577	0.315	−0.347	0.497	0.313	0.395	0.863	0.483	−0.883	0.501	0.485	0.452	1			
**Pb**	−0.519	−0.249	0.410	−0.436	−0.247	−0.331	−0.826	−0.422	0.848	−0.441	−0.423	−0.389	−0.998 *	1		
**V**	−0.429	−0.148	0.502	−0.341	−0.146	−0.232	−0.764	−0.327	0.790	−0.346	−0.328	−0.293	−0.985	0.995	1	
**Zn**	−0.173	0.120	0.714	−0.079	0.122	0.035	−0.564	−0.063	0.598	−0.084	−0.065	−0.028	−0.904	0.932	0.964	1

* Correlation is significant at the 0.05 level. ** Correlation is significant at the 0.01 level.

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
