# Peer review of "Characterization of the 12S rRNA Gene Sequences of the Harvester Termite Anacanthotermes ochraceus (Blattodea: Hodotermitidae) and Its Role as A Bioindicator of Heavy Metal Accumulation Risks in Saudi Arabia"

_insects, 2019, doi:10.3390/insects10020051_

Round 1

Reviewer 1 Report

This paper seems like two distinct papers that were put together. One paper describes a phylogenetic analysis of the termites and suggests some realignments of current classification. This is an interesting and well done study. Using termites as a bioindicator of heavy metal pollution is a separate and seemingly unrelated topic. Again, this study is well done and informative. I do not really see how the two studies relate and why they should be combined in a single paper. Can the authors justify the combination of the two topics? If not, I would suggest splitting the paper.

The manuscript does not conform to journal format. Specifically, the references are not cites correctly in the text; consecutive numbers should be used. Also, the individual references in the reference section are not formatted correctly.

The authors need to consistently italicize the scientific name of their species whether the genus is spelled out or abbreviated: Anacanthotermes ochraceus or A. ochraceus throughout the manuscript.

I have attached a pdf of the manuscript that contains my edits, suggestions, and questions.

Author Response

This paper seems like two distinct papers that were put together. One paper describes a phylogenetic analysis of the termites and suggests some realignments of current classification. This is an interesting and well done study. Using termites as a bioindicator of heavy metal pollution is a separate and seemingly unrelated topic. Again, this study is well done and informative. I do not really see how the two studies relate and why they should be combined in a single paper. Can the authors justify the combination of the two topics? If not, I would suggest splitting the paper.

….. Many thanks for your positive feedback about our MS,

….. The reason for combination the both topic in this MS is “The main purpose of the MS is to assess the ability of the recovered termite species as a biological indicator for the degree of environmental pollution in the studied area. While, firstly we need to make accurate determination for its type… for that, we make molecular identification firstly then the test of measurements of heavy metals”

In addition we added these briefly in abstract as “Heavy metal accumulation causes serious health problems in humans and animals. Being involved in the food chain, insects are used as bioindicators of heavy metals. In the present study, 100 termite individuals of Anacanthotermes ochraceus were collected from two Saudi Arabian localities with different geoclimatic conditions (Riyadh and Al Taif). These individuals were subjected to morphological identification followed by molecular analysis using mitochondrial 12S rRNA gene sequence, thus confirming the morphological identification of A. ochraceus.” Then described in detail in the whole text.

Also, in the aim of the MS was “Therefore, the present study aimed to determine the morphological and molecular characterizations of the harvester termite Anacanthotermes ochraceus in order to determine its taxonomic position within the family Hodotermitidae and to assess whether this hodotermitid species could be used as a candidate bio-indicator of soil heavy metals.”

The manuscript does not conform to journal format. Specifically, the references are not cites correctly in the text; consecutive numbers should be used. Also, the individual references in the reference section are not formatted correctly.

….. OK, it appropriate and done

The authors need to consistently italicize the scientific name of their species whether the genus is spelled out or abbreviated: Anacanthotermes ochraceus or A. ochraceus throughout the manuscript.

….. OK, it appropriate and done

I have attached a pdf of the manuscript that contains my edits, suggestions, and questions.

….. OK, it appropriate and done for editing and suggestions

….. For questions >>>> as following

The order Isoptera must be changed into Order Blattodea

…… It is appropriate and done

The number of families must be changed from seven into twelve

…… It is appropriate and changed from “seven families and 14 subfamilies (Engel and Krishna, 2004)” into “12 families and 14 subfamilies (Krishna et al. 2013)”

In Materials and Methods

What are the geographical coordinates for the studied sites?

….. OK, it appropriate and changed to be “Two distinct natural habitats in Saudi Arabia were selected for the current investigation: site I at An-Nazim District, Eastern region of Riyadh City (24°41'15.83"N, 46°43'18.66"E) and site II at Arfaa District, Northern region of Taif City (21° 16' 36.2892'' N, 40° 22' 28.1172'' E).”

How were they collected? Traps or indexed wood?

…… It is appropriate and changed from “Anacanthotermes ochraceus were randomly collected with sand” into “Anacanthotermes ochraceus were randomly collected from indexed wood and sand of two”

In Table 1. The word Blattoidea must change into Blattodea

….. OK, it appropriate and changed

In Figure 2.. All names of termites must be italic

….. OK, it appropriate and changed

Write the equation DC=∑_(i=1)^n▒CFi in right form

….. OK, it appropriate and changed into DC=

For Statitical analysis : Duncan’s multiple range tests …. Use another test

….. It removed from the text

In Results

About Rare hairs?

….. It changed into tiny hairs

In chemical analysis >>>> move sentence into the discussion of “The concentrations of these contaminants were found in the hazardous levels that exceeded the permissible limits for mineral soil recommended by the standards of the Egyptian High Committee of Water (EHCW, 1995), Saudi Standards Metrology and Quality Organization (SASO, 1997), and United States Environmental Protection Agency (USEPA, 1998).”

….. OK, it appropriate and moved

How progression >>>> did you sample one, time?

….. This sentence “that showed a progressive deterioration of the site quality. Furthermore, the contamination degree of Site II indicates very high degree of pollution” was removed from the text

Reviewer 2 Report

Dear authors,

The suggestions and comments can be found in the attached file. The manuscript is well written, however it is necessary to update the references.

Kind regards,

Author Response

The suggestions and comments can be found in the attached file. The manuscript is well written, however it is necessary to update the references.

….. Many thanks for your positive feedback about our MS,

….. OK, all the required points are done in appropriate way

As following:

In whole Text : Anacanthotermes ochraceus …. Be italic

….. OK, it appropriate and done

In whole Text : Reference must be numbered

….. OK, it appropriate and done

In Keywords : Isoptera; Hodotermitidae …. Must be deleted as it found on the title

….. OK, it appropriate and changed into Blattodea

In Introduction : Isoptera must be changed by Blattodea

 ….. OK, it appropriate and changed

In Materials and Methods : What are the geographical coordinates for the studied sites?

….. OK, it appropriate and changed to be “Two distinct natural habitats in Saudi Arabia were selected for the current investigation: site I at An-Nazim District, Eastern region of Riyadh City (24°41'15.83"N, 46°43'18.66"E) and site II at Arfaa District, Northern region of Taif City (21° 16' 36.2892'' N, 40° 22' 28.1172'' E).”

In Materials and Methods : Murthy et al. (2015) is missed in the reference list

….. OK, it appropriate and added in the reference list as “Murthy, S.; Rajeshwari, R.K.; Ramya, S.L.; Venkatesan, T.; Jalali, S.K.; Verghese, A. Genetic diversity among Indian termites based on mitochondrial 12S rRNA gene. European Journal of Zoological Research 2015, 4 (1), 1-6.”

In Materials and Methods : In Table (1) Blattoidea must be changed into Blattodea

….. OK, it appropriate and changed

In Results : Schedorhinotermes, Drepanotermes, Macrognathotermes, Nasutitermes, Anacanthotermes, Hodotermes, Microhodotermes…. All must be italic

….. OK, it appropriate and changed

In Results : The word order Isoptera must be changed into infraorder Isoptera

….. OK, it appropriate and changed

In Discussion : the word Isoptera must be changed by Blattodea

….. OK, it appropriate and changed

In Discussion : Okhuma et al. (2004) mentioned and not found in reference list

….. OK, it appropriate and added in the reference list as “Ohkuma, M.; Yuzawa, H.; Amornsak, W.; Sorrnuwat, Y.; Takematsu, Y.; Yamada, A.; Vongkaluang, C.; Sarnthoy, O.; Kirtibutr, N.; Noparatnaraporn, N.; Kudo, T.; Inoue, T. Molecular phylogeny of Asian termites (Isoptera) of the families Termitidae and Rhinotermitidae based on mitochondrial COII sequences. Mol Phylogenet Evol 2004, 31, 701-710”

In Discussion : Ahmed (1950) mentioned and not found in reference list

….. OK, it appropriate and added in the reference list as “Ahmad, M. The phylogeny of termite genera based on imagoworker mandibles. Bulletin of the American Museum of Natural History 1950, 95, 37-86.”

In Discussion : Why the word Blattodea not mentioned in the text

….. OK, it appropriate and mentioned in the text

In Discussion : The word Mastotermes, Heterotermes, Naustitermes must be italic

….. OK, it appropriate and done

In Discussion : Pnnanen et al. (1996a,b) mentioned and not found in reference list

….. OK, it appropriate and added in the reference list as “Pennanen, T.; Fostegråd, Å.; Fritze, H.; Bååth, E. Phospholipid fatty acid composition and heavy metal-polluted gradients in coniferous forests. Applied Environmental Microbiology 1996a, 62, 420-428.

Pennanen, T.; Frostegård, Å.; Fritze, H.; Bååth, E. Phospholipid fatty acid composition and heavy metal tolerance of soil microbial communities along two heavy metal-pollution gradients in coniferous forests. Applied and Environmental Microbiology 1996b, 62, 420-428.”

In Discussion : Grazesiak and Sieradzki (2000) mentioned and not found in reference list

….. OK, it appropriate and added in the reference list as “Grzesiak, M.; Sieradzki, Z. Environment. Information and statistical papers. Statistical Publications House, Warsaw, 2000.”

In Reference : all of them must be as Journal formats

….. OK, it appropriate and added

Round 2

Reviewer 1 Report

This revision is well done and reads very clearly. I am still somewhat concerned about how the two projects (termite taxonomy and heavy metal accumulations) are put together. I think using molecular methods to correctly identify the termite species that was used for heavy metal research could be a single paper; a second paper on phylogenetics of termites could stand alone.

That being said, the paper as written is certainly acceptable. Good ideas and execution.